# A Virtual Combustion Sensor Based on Ion Current for Lean-Burn Natural Gas Engine

**DOI:** 10.3390/s22134660

**Published:** 2022-06-21

**Authors:** Xiaoyan Wang, Tanqing Zhou, Quan Dong, Zhaolin Cheng, Xiyu Yang

**Affiliations:** 1Institute of Power and Energy Engineering, Harbin Engineering University, Harbin 150001, China; wang_xiaoyan@hrbeu.edu.cn (X.W.); dong_quan@hrbeu.edu.cn (Q.D.); zhaolinCheng@hrbeu.edu.cn (Z.C.); yangxiyu@hrbeu.edu.cn (X.Y.); 2Weichai Power Co., Ltd., Weifang 261061, China

**Keywords:** combustion sensor, ion current, online measurement, neural network

## Abstract

In this study, an innovative sensor was designed to detect the key combustion parameters of the marine natural gas engine. Based on the ion current, any engine structurally modified was avoided and the real-time monitoring for the combustion process was realized. For the general applicability of the proposed sensor, the ion current generated by a high-energy ignition system was acquired in a wide operating range of the engine. It was found that engine load, excess air coefficient (λ) and ignition timing all generated great influence on both the chemical and thermal phases, which indicated that the ion current was highly correlated with the combustion process in the cylinder. Furthermore, the correlations between the 5 ion current-related parameters and the 10 combustion-related parameters were analyzed in detail. The results showed that most correlation coefficients were relatively high. Based on the aforementioned high correlation, the novel sensor used an on-line algorithm at the basis of neural network models. The models took the characteristic values extracted from the ion current as the inputs and the key combustion parameters as the outputs to realize the online combustion sensing. Four neural network models were established according to the existence of the thermal phase peak of the ion current and two different network structures (BP and RBF). Finally, the predicted values of the four models were compared with the experimental values. The results showed that the BP (with thermal) model had the highest prediction accuracy of phase parameters and amplitude parameters of combustion. Meanwhile, RBF (with thermal) model had the highest prediction accuracy of emission parameters. The mean absolute percentage errors (MAPE) were mostly lower than 0.25, which proved a high accuracy of the proposed ion current-based virtual sensor for detecting the key combustion parameters.

## 1. Introduction

Increasingly stringent regulations on engine emissions have made alternative fuels a hot topic [1]. Natural gas has become one of the main alternative fuels for the internal combustion engine due to its low hydrocarbon ratio and renewable properties [2,3]. Moreover, the high-octane number and high antiknock properties of natural gas enable the engine to run at a higher compression ratio, thus improving thermal efficiency. However, there are some non-negligible obstacles to the development of lean-burn natural gas engines. For example, the gas in the cylinder is lean so that higher ignition energy must be used, which will cause the deterioration of the durability of the ignition system. Furthermore, when the gas mixture is thin enough, combustion becomes extremely unstable, and even causes the misfire phenomenon. It not only reduces engine efficiency, but also increases HC emissions. To solve the problems mentioned above, realizing on-line combustion sensing is the primary task.

At present, there are two main methods to realize combustion sensing. One method is to use optical testing technology to analyze the combustion condition of the constant volume projectile. In the past few decades, laser-based combustion detection technology has been developed and has become a recognized method of combustion sensing. Researchers used the schlieren method, PIV, LIF and other optical testing methods to obtain key information about the working process in the engine cylinder, such as the temperature field, concentration distribution of each component in the combustion field, flame characteristics, soot generation characteristics, etc. [4,5]. This method, however, is mostly used on the laboratory bench. In fact, the complexity of the actual engine structure makes optical testing difficult and it cannot meet the requirements of real-time monitoring of the combustion process. The other method is to install the cylinder pressure sensor on the cylinder cover. By processing the measured in-cylinder pressure signal, not only peak pressure, average pressure and other parameters can be obtained directly, but also the heat release rate and the combustion reaction extent can be calculated, which are of great importance to engine structure design and combustion diagnosis [6,7,8,9]. However, due to harsh operating conditions, the in-cylinder pressure sensor has a short duration, which greatly increases the cost of engine production. In contrast, combustion sensing based on ion current only needs to transform the spark plug into a combustion status sensor, which is simple and low-cost. Furthermore, the abundant combustion information contained in the ion current is sufficient to realize the combustion analysis [10,11,12,13,14]. Therefore, the ion current sensor has the potential to replace the in-cylinder pressure sensor.

A large number of studies have shown that the spark plug ion current signal contains abundant combustion information and can reflect the combustion situation in the cylinder. Andersson built a model for the thermal part of an ionization signal in a four stroke SI (spark ignition) engine, which can be used to estimate combustion properties such as pressure, temperature, and content of nitric oxides based on measured ionization currents with good accuracy [15]. Gerard proposed a methodology based on ion current for extraction of critical parameters including combustion phasing, knock detection and combustion stability [16]. Liu et al. applied the ion current to HCCI combustion sensing and proposed a thermal phase model of the HCCI combustion ion current signal, which was used to analyze the relationship between combustion parameters and ion current parameters [17]. Furthermore, the combustion signal sensing technology based on ion current has been introduced into the study of fuel ratio of ethanol/gasoline dual fuel engine.

Without any engine modification, the spark plug ion current method has great potential in engine combustion perception research. However, there are still some disadvantages. Firstly, the ion current signal is very weak and vulnerable to electromagnetic interference and environmental interference from the ignition system [18]. Secondly, the ion current signal is related to the engine structure, the strength of the ion current signal is affected by the spark plug structure, fuel properties and the shape of the combustion chamber. In addition, the information about the combustion process contained in the ion current needs to be preprocessed before it can be used for engine control. So far, there is no unified opinion about how to extract the information about combustion contained in the ion current accurately and quickly. This is because the ion current presents different characteristics in different types of engines with different combustion strategies [19,20].

At present, most of the research on ion current is focused on the gasoline engine [21,22,23], but less so on the natural gas engine, resulting in few available reference data. Therefore, this paper aims at the ion current in the lean-burn natural gas engine to provide data support for further investigation. To fully study the characteristics of the ion current, the main factors affecting the chemical phase and thermal phase of the ion current under lean burn combustion conditions were investigated, including engine load, excess air coefficient (λ) and ignition timing. Then, the correlation between phase and amplitude characteristics of the ion current and the combustion characteristics was analyzed. Furthermore, the BP and RBF neural networks were trained by the experimental value of whether there was an ion current thermal phase, and four sensing models of combustion sensing based on the ion current signal were obtained accordingly. Finally, the accuracy of the models was evaluated and compared. The results were applied to the electronic control unit.

## 2. Experimental Setup

The experiments were carried out on a natural gas engine. The specifications of the engine are presented in Table 1, and an overview of the experimental set-up is shown in Figure 1.

The engine was tested using a GW500 dynamometer. On one of the engine cylinders, the spark plug served as an ion current sensor, and a Kistler 4067 pressure sensor was applied to measure the in-cylinder pressure. The signal was amplified and transmitted (sampled every 0.1 crank angles) to a Ki-Box 2893A combustion analyzer. Meanwhile, an AMA i60 emission analyzer was employed to measure NOx emissions transiently. More importantly, a Lambda Meter LA4 air–fuel ratio analyzer was employed to indicate the mixture concentration.

To guarantee the high signal to noise ratio of the ion current under lean-burn condition, the NGI-1000 high energy ignition system developed by Altronic company was employed to replace the original ignition system. The high ignition system enables the modification of both ignition energy and ignition duration, which improves combustion stability in lean operating environments. The system consists of an Altronic 502061 ignition coil, a controller and connecting cables. The NGI-1000 unit steps up a 24V DC supply voltage to charge an energy storage capacitor, and the voltage of the primary coil is 185V. The ion current measurement circuit is displayed in Figure 2.

To synchronize ion current with crank angle signal, the ion current was acquired by the Ki-Box 2893A combustion analyzer through a voltage signal acquisition channel. Thus, the ion current, the crank angle and the in-cylinder pressure were all measured synchronically. A 300 Ω measuring resistance was employed to obtain the ion current signal practically. A cycle of data is plotted in Figure 3.

The fuel combustion process is accompanied by a large number of chemical reactions, resulting in a large number of ions. At this time, the applied bias voltage can make the ions near the spark plug electrode move oriented to generate a current, which is called the ion current. In the ignition phase, the current is the jump current formed by the discharge of the ignition coil in the spark plug gap. In the flame-front phase, the current produced by the ion surge resulting from the chemical reaction of flame combustion is called the chemical phase ion current. In the post-flame phase, ionization of the high-temperature gas mixture after combustion causes the thermal phase ion current.

Five characteristics related to ion current are defined in this paper. Chemical peak value (CPV) indicates the peak value of chemical phase ion current, and chemical peak phase (CPP) is the corresponding crank angle. Similarly, thermal peak value (TPV) indicates the peak value of thermal phase ion current, and thermal peak phase (TPP) is the corresponding crank angle. In addition, ion current integration (ICI) stands for the integration of both chemical and thermal ion current. The formula of ICI is
(1)$$\mathrm{ICI}=\int_{t_{1}}^{t_{2}}i(t)dt$$
where *t*_1_ is the time corresponding to the crank angle when the ion current changes from negative to positive, *t*_2_ is the time corresponding to crankshaft rotation when the amplitude of ion current decreases to 0, *i*(*t*) is the time domain signal of the ionic current.

To fully study the influencing factors of ion current and the correlation between ion current characteristics and combustion characteristics, all test conditions are listed as shown in Table 2.

Where X is the excess air coefficient under the propulsion characteristics of the original machine under the current load. In the actual experiment, the engine in some working conditions was unstable, so the data in these working conditions were not used. In particular, when the load was 25, 50, 75 and 100%, besides collecting ion current, in-cylinder pressure signal, excess air coefficient and other engine operating parameters, emission data were collected in parallel.

## 3. Results

### 3.1. Ion Current Signals under Different Operating Conditions

#### 3.1.1. Excess Air Coefficient Effects on Ion Current Signals

As one of the most important factors affecting engine combustion, λ (excess air coefficient) directly affects the number of molecules involved in the combustion reaction and ionization, which has a significant impact on the ion current. To evaluate this effect, the ignition timing was maintained at 37 °CA before top dead center (BTDC) under 50% engine load, and the excess air coefficient was adjusted by changing the injection pulse width and throttle opening [24]. Figure 4 shows the average ionic current signals of 140 cycles with excess air coefficient between 1.3 and 1.6.

In Figure 4, with the increase of the excess air coefficient, the overall signal strength of the ion current weakened. Both the CPV and the TPV decreased significantly with the increase of the excess air coefficient, and the decrease range of TPV was larger than that of CPV. Furthermore, when the excess air coefficient exceeded 1.50, the TPV even disappeared. The CPP and TPP were delayed with the increase of the excess air coefficient.

#### 3.1.2. Ignition Timing Effects on Ion Current Signals

Ignition timing affects the flame propagation rate and ionization ratio of the mixture by affecting the temperature of the combustible mixture in the cylinder, thus affecting the local peak value and phase of the ion current. Therefore, in this paper, the excess air coefficient was kept unchanged at 1.35 under 50% engine load. The effect of ignition timing on the ion current was studied by changing the ignition timing of NGI-1000 ignition system. As shown in Figure 5, the average ionic current signals when the ignition timing was 25, 28, 31, 34, 37 and 40 °CA BTDC are depicted.

As the ignition timing moved forward, the overall ionic current signal intensity gradually increased. Both the CPV and TPV increased significantly with the advance of ignition time, and the increase of thermal TPV was larger than that of CPV. The CPP and the TPP of the ion current moved forward with the ignition timing, and the peak phase of the chemical phase moved forward more obviously.

#### 3.1.3. Engine Load Effects on Ion Current Signals

Figure 6a–c show the relationship between CPV, TPV, ICI and BTDC, respectively, as well as engine load when the excess air coefficient was kept at 1.35. CPV increased first and then decreased with the increase of load, and it increased gradually with the advance of ignition timing. In general, TPV increased with the increase of load and the advance of ignition timing. ICI is the integral of ionic current in the whole process, and the influence of BTDC and load on ICI can be seen as the superposition of its action on CPV and TPV. As can be seen from (a) and (b), the influence of BTDC and load on TPV was more significant than that of CPV, so the influence of BTDC and load on ICI was closer to TPV.

### 3.2. Correlation of Ion Current with Combustion and Emission Parameters

Figure 7 and Figure 8 shows the relationship between the ion current phase parameters (CPP and TPP) and the combustion phase parameters (CA05, CA50, CA90, Pmax(θ), γ(θ) and dPmax(θ)) under all load conditions. Generally, CA05 is the phase when the cumulative heat release rate reaches 5%, which is used to indicate the starting point of combustion. CA50 is the phase where the cumulative heat release reaches 50%. CA90 is the phase with a cumulative heat release rate of 90%, which is generally used to indicate the end point of combustion during combustion. Pmax(θ) is the phase corresponding to the maximum value of the in-cylinder pressure curve. γmax(θ) is the phase corresponding to the maximum heat release curve. dPmax(θ) is the phase corresponding to the maximum value of derivative curve of in-cylinder pressure.

As depicted in Figure 7, the discrete points represent the data under all working conditions with excess air coefficients of 1.3, 1.4 and 1.5, respectively. CPP had a high linear correlation with CA05, CA50, CA90, Pmax(θ), γ(θ) under the same excess air coefficient. Furthermore, the correlation coefficients under the three different excess air coefficients were mostly more than 0.9.

As shown in Figure 8, the discrete points represent the combustion phase parameters under all working conditions without differentiation of excess air coefficient. The TPP was highly correlated with all six combustion phase characteristics. CA05, CA50, CA90, CPP, Pmax(θ) and γ Max (θ) under three different excess air coefficients were mostly more than 0.87.

As shown in Table 3, both the CPP and TPP can better reflect the characteristics of combustion, while using the CPP to reflect the combustion phase parameters needed an excessive air coefficient as an additional condition. As a result, CA05, CA50, CA90, Pmax(θ) and γ(θ) could be indicated by the CPP combined with excess air coefficient or the TPP alone. Furthermore, dPmax(θ) could only be indicated by the TPP alone.

The in-cylinder pressure can directly reflect the combustion state in the cylinder, Pmax is the maximum pressure in the combustion process and combined with the peak of the heat release curve γmax, can be used to indicate the sufficient degree of the combustion process. Figure 9 shows the correlation between Pmax, dPmax, γmax and CPV, TPV, ICI in all working conditions under a 75% load.

The correlation between the Pmax and the three amplitude characteristics of the ion current was weak. This is because the pressure in the cylinder is not only related to the combustion process in the cylinder, but also affected by the compression pressure generated by the piston movement. On the other hand, dPmax had a high correlation with the amplitude of the ion current. The relationship between the three combustion parameters and ICI was more logarithmic. γ max was highly linearly correlated with CPV, and more logarithmic with TPV and ICI. As a result, dPmax and γ max can be reflected by the amplitude characteristics of ion current.

NOx emission data were measured at 25, 50, 75 and 100% load conditions. In order to ensure the accuracy of the data, the data under each working condition was recorded after a stabilization time of one minute. As depicted in Figure 10, with the increase of CPV, NOx emission increased. Although CPP has a certain correlation with NOx emissions, it cannot fully reflect NOx emissions. For TPV, NOx emissions showed a rising trend with the increase of TPV, and TPV is highly correlated with NOx emissions, showing a logarithmic correlation. That is, when TPV is small, NOx emissions increase rapidly, while when TPV is large, NOx emissions increase slowly. Similarly, compared with linear fitting data points, the relationship between ICI and NOx emissions is more consistent with the logarithmic fitting trend line.

To conclude, the engine operating conditions determined great effects on the ion current. The excess air coefficient affects the amplitude of the ion current during the whole stage, especially the thermal phase stage. When λ varied from 1.3 to 1.6, the TPV varied from 0 to 15 mA. The ignition timing affected both the amplitude and the phase of the ion current. Besides, the CPV, TPV, ICI were sensitive to the change of engine load. More specifically, the correlation between the ion current and the combustion parameters were relatively high. Therefore, the ion current could be used to realize combustion sensing [25,26,27]. An ion-current based virtual combustion sensor is proposed as below.

## 4. Virtual Combustion Sensor Based on Ion Current

### 4.1. ANN Neural Network Models

As a global approximation network, the back-propagation (BP) neural network has a slow learning speed and is not suitable for applications requiring high real-time performance. In contrast, the radial basis function (RBF) network has the characteristics of simple structure, fast learning speed and good function fitting performance and has been applied in different industries and fields and performed well. Therefore, this paper uses the BP neural network algorithm and RBF neural network to propose a virtual combustion sensor based on ion current [28,29,30]. The number of the input nodes and output nodes of the networks are the same, while the number of hidden nodes are different. As for the BP neural network, the number of the hidden nodes are determined by commonly used formula. For the RBF neural network, hidden nodes are added to the hidden layer until the mean square deviation meets the requirements. The network topology is shown in Figure 11.

The combustion sensor consists of four combustion parameter-related models, which are the excess air coefficient prediction model, combustion phase model, combustion amplitude model and NOx emission model. Their inputs and outputs are shown in Figure 12 below.

The experimental data of the natural gas engine are divided into two groups according to the presence or absence of the ion current thermal phase, and randomly shuffled. Then, 70% of the samples from 388 samples (single flow thermal phase with ion) and 127 samples (single flow thermal phase without ion) were taken to train the models. Finally, 20 samples were taken as test and validation samples to verify the accuracy of the established virtual sensor.

### 4.2. Evaluation of the Prediction Performance

After training, the output of networks needs to be compared with the experimental data tested by practical devices to verify the performance of the sensor. To evaluate the accuracy, root mean square error (RMSE), tie relative error (MAPE) and relative error are applied. The formulas are as follow:(2)$$\mathrm{RMSE}=\sqrt{\frac{1}{n}\sum_{i=1}^{n}{\left(f_{i}-m_{i}\right)}^{2}}$$
(3)$$\mathrm{MAPE}=\frac{1}{n}\sum_{i=1}^{n}\left(\frac{\left|f_{i}-m_{i}\right|}{m_{i}}\right)$$
(4)$$\mathrm{RE}=\left|\frac{f_{i}-m_{i}}{m_{i}}\right|$$
where *m_i_* is the test value, *f_i_* is the predicted value, and *n* is the number of test samples.

For example, the prediction results of λ models (both the one with thermal phase and the one without thermal phase) are shown in the Figure 13 and Figure 14.

As can be seen from Figure 13, among the models containing the thermal phase data of the ion current, both BP and RBF models could well predict the excess air coefficient. The residual between the predicted value of the network and the experimental value was within 0.05. Figure 14 illustrates that in the model without thermal phase data of ionic current, the errors of BP and RBF models were slightly larger. The model with thermal phase ion current showed higher accuracy.

As depicted in Figure 15, for each combustion phase model, no matter whether BP or RBF network model, the RMSE value of the model containing the thermal phase characteristics of ion current was generally lower than that of the model without the ion current thermal phase, and the value was less than 1. This shows that the neural network model containing the thermal phase characteristics of the ion current can predict each combustion phase with high accuracy. In the neural network model without thermal phase characteristics of ion current, only CA05 and the maximum in-cylinder pressure phase model had low RMSE values, while other combustion phase models all had RMSE values greater than 1. This shows that the neural network model without the thermal phase characteristics of the ion current can only predict CA05 and the maximum pressure phase with high accuracy. In the neural network models containing the thermal phase characteristics of the ion current, BP neural network models tended to have lower RMSE and MAPE values, except for the maximum heat release rate model. In CA05 and maximum in-cylinder pressure phase prediction models without thermal phase characteristics of ion current, RBF network model RMSE and MAPE values were low and had higher prediction accuracy.

As can be seen from Figure 15, similar to the combustion phase model, both BP and RBF network models had lower RMSE and MAPE values than those without ionic current thermal phase characteristics. It can be seen that ionic current thermal phase characteristics can significantly increase the prediction accuracy of a neural network model. In the neural network model without the thermal phase characteristics of the ion current, the MAPE values of the maximum in-cylinder pressure model and the maximum combustion heat release rate were lower, but both were greater than 5%. The RMSE and MAPE values of BP neural network were higher than those of RBF neural network. This indicates that the prediction accuracy of the neural network model without the thermal phase characteristics of the ion current is limited, and the accuracy of the BP network model is lower than that of the RBF network model. In the neural network model containing thermal phase characteristics of ion current, the BP neural network model had lower RMSE and MAPE values. The average relative errors of the two networks for predicting the maximum in-cylinder pressure and the maximum combustion heat release rate were less than 5%, and the average relative errors for predicting the maximum in-cylinder pressure increase rate were slightly higher, 5.16% and 5.57% respectively. The model still maintained high accuracy.

For NOx emission, the prediction accuracy of the model containing the thermal phase characteristics of the ion current was much better than that of the model without the thermal phase characteristics of the ion current, and the MAPE values of the predicted results were all lower than 0.1. As shown in Figure 16, the RMSE value of the model without ionic current thermal phase characteristics was small by contrast. The reason is that the samples without ionic current thermal phase were measured under the condition of low gas concentration, and the NOx emission was small.

Therefore, in this case, the RMSE value could not reflect the accuracy of the two models. The accuracy of the BP network model was lower than that of the RBF model in the model containing the thermal phase characteristics of the ion current, but it was the opposite in the model without the thermal phase characteristics of the ion current.

## 5. Conclusions

The characteristic parameters of the ion current are related to the natural gas engine operating conditions. The excess air coefficient affects the amplitude of the ion current during the whole stage, especially the thermal phase stage. When λ varies from 1.3 to 1.6, the TPV varies from 0 to 15 mA. The ignition timing affects both the amplitude and the phase of the ion current. Furthermore, CPV, TPV, ICI are sensitive to the change of engine load.The correlation between the ion current characteristics and the combustion characteristics are relatively high. The correlation coefficients are mostly higher than 0.85.The virtual sensor based on the ion current has good prediction results for excess air coefficient, combustion phase parameters, combustion amplitude parameters and NOx emission. For the prediction of excess air coefficient, combustion phase parameters and combustion peak, the accuracy of the BP neural network is higher than that of RBF when there is a thermal phase in an ion current signal. When the thermal phase of the ion current disappears, RBF shows a higher prediction accuracy.

## Figures and Tables

**Figure 1 sensors-22-04660-f001:**
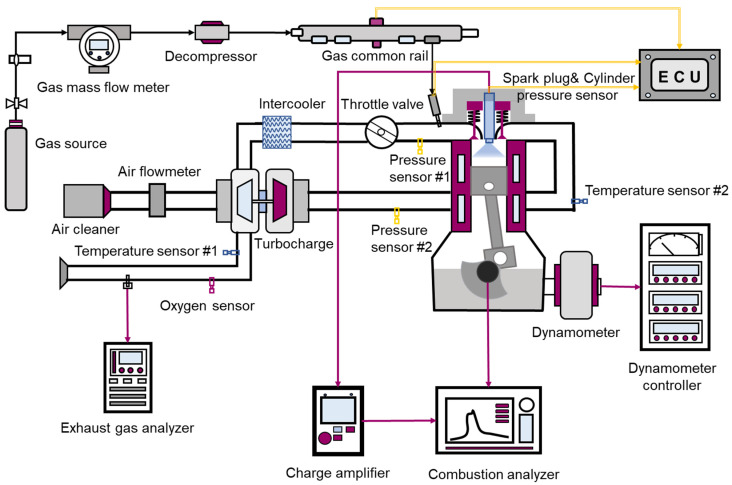
Diagram of test bench installation.

**Figure 2 sensors-22-04660-f002:**
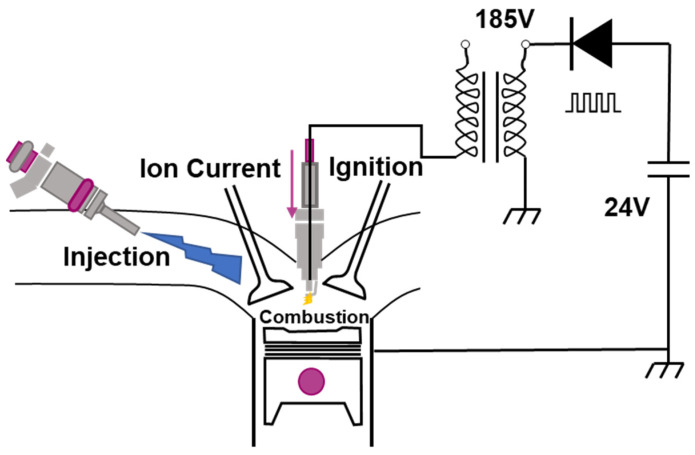
Circuit diagram of the ion current measurement.

**Figure 3 sensors-22-04660-f003:**
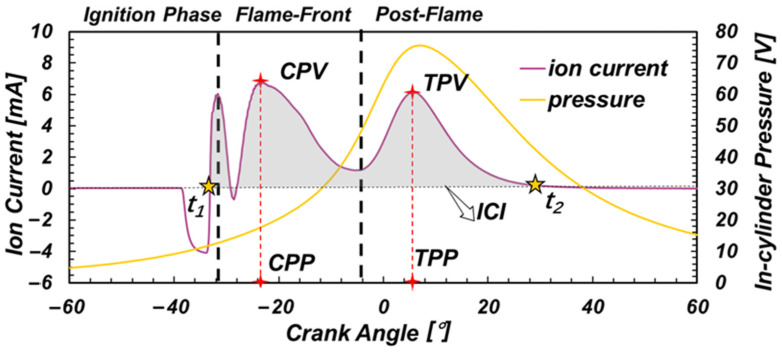
The ion current and in-cylinder pressure.

**Figure 4 sensors-22-04660-f004:**
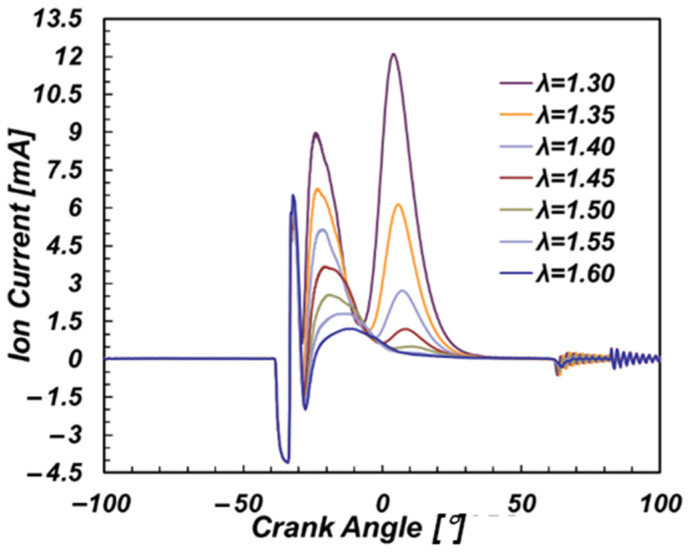
Effect of excess air coefficient on ionic current.

**Figure 5 sensors-22-04660-f005:**
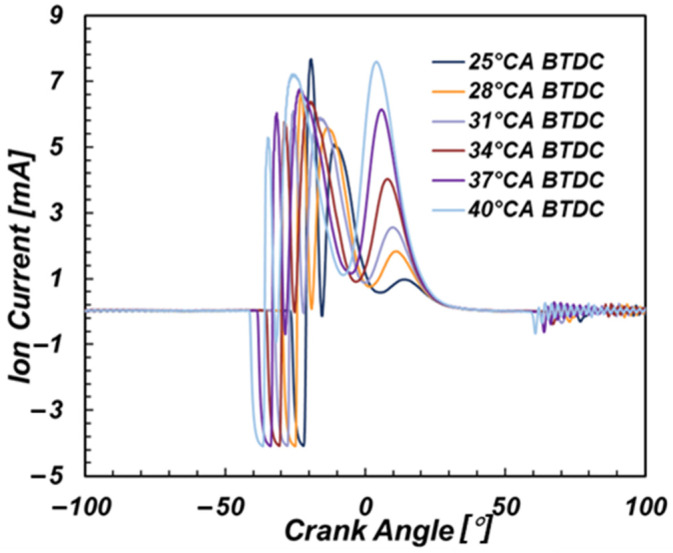
Effect of ignition timing on ionic current.

**Figure 6 sensors-22-04660-f006:**
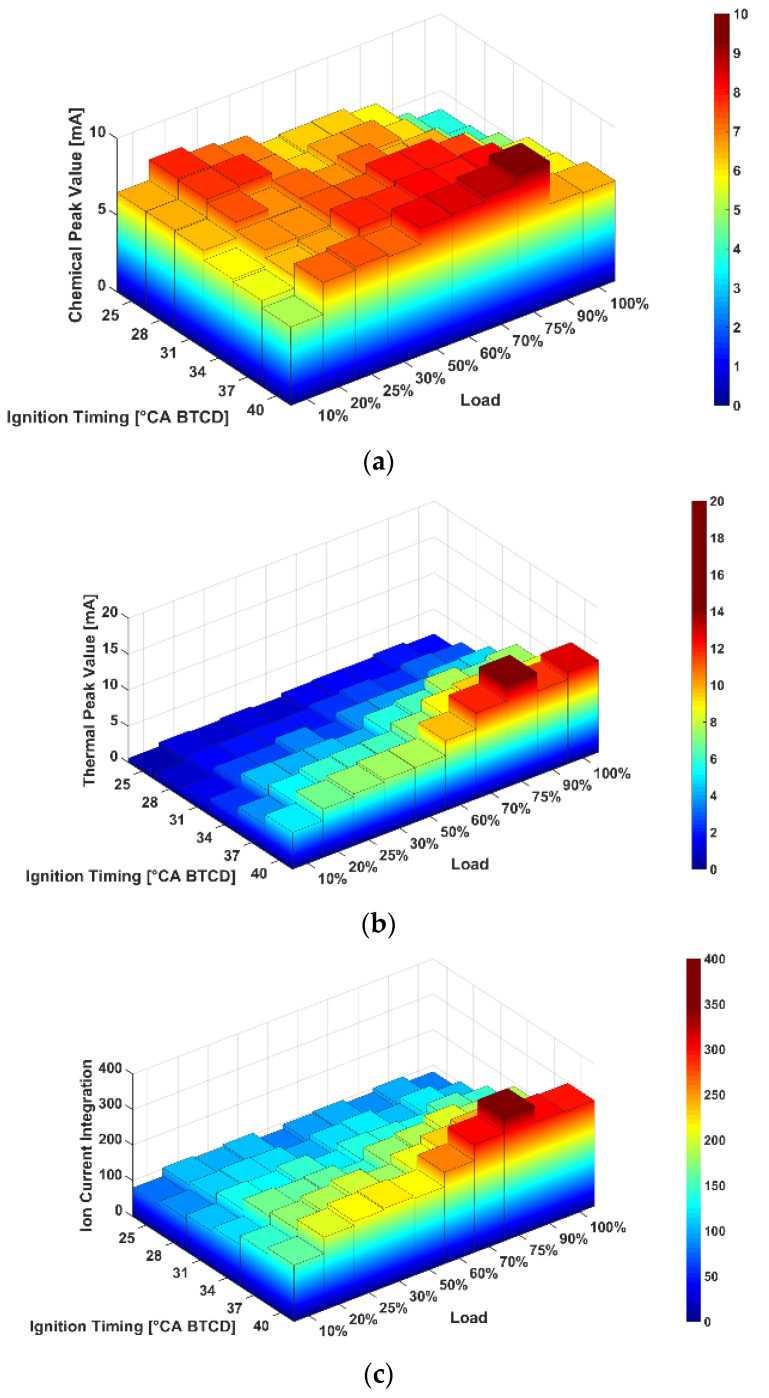
The influence of load and ignition timing on ion current. (**a**) Influence on CPV; (**b**) Influence on TPV; (**c**) Influence on ICI.

**Figure 7 sensors-22-04660-f007:**
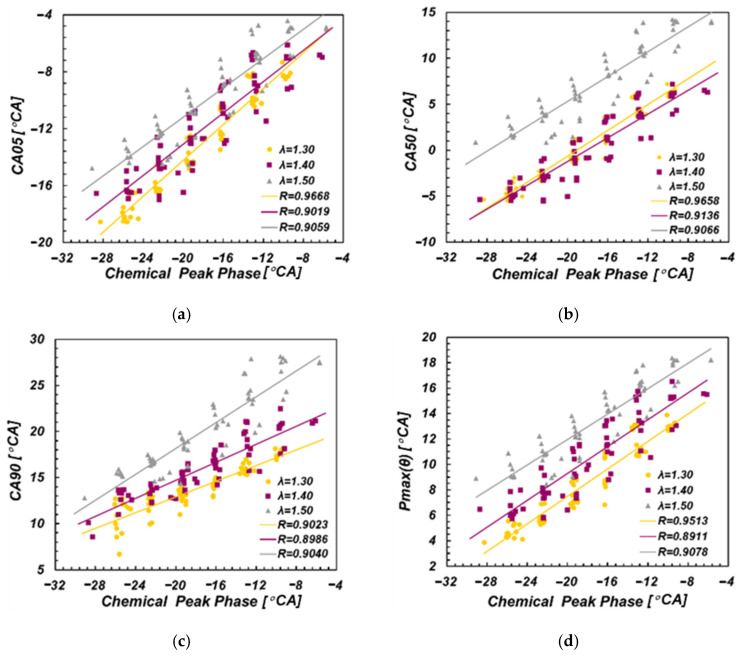
Correlation between CPP and combustion phase. (**a**) CA05; (**b**) CA50; (**c**) CA90; (**d**) Pmax(θ); (**e**) γmax(θ); (**f**) dPmax(θ).

**Figure 8 sensors-22-04660-f008:**
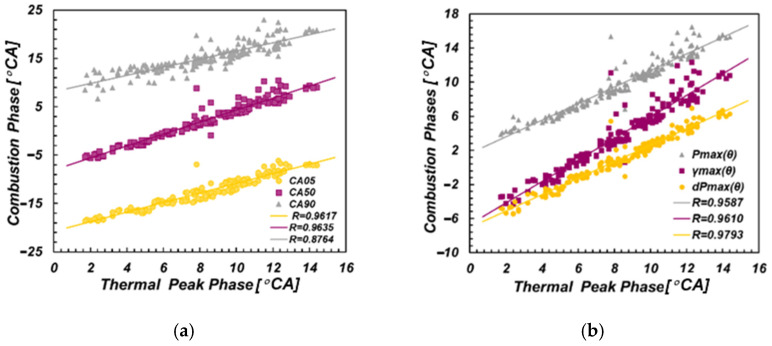
Correlation between TPP and combustion phase. (**a**) CA05, CA50 and CA90; (**b**) Pmax(θ), γmax(θ) and dPmax(θ).

**Figure 9 sensors-22-04660-f009:**
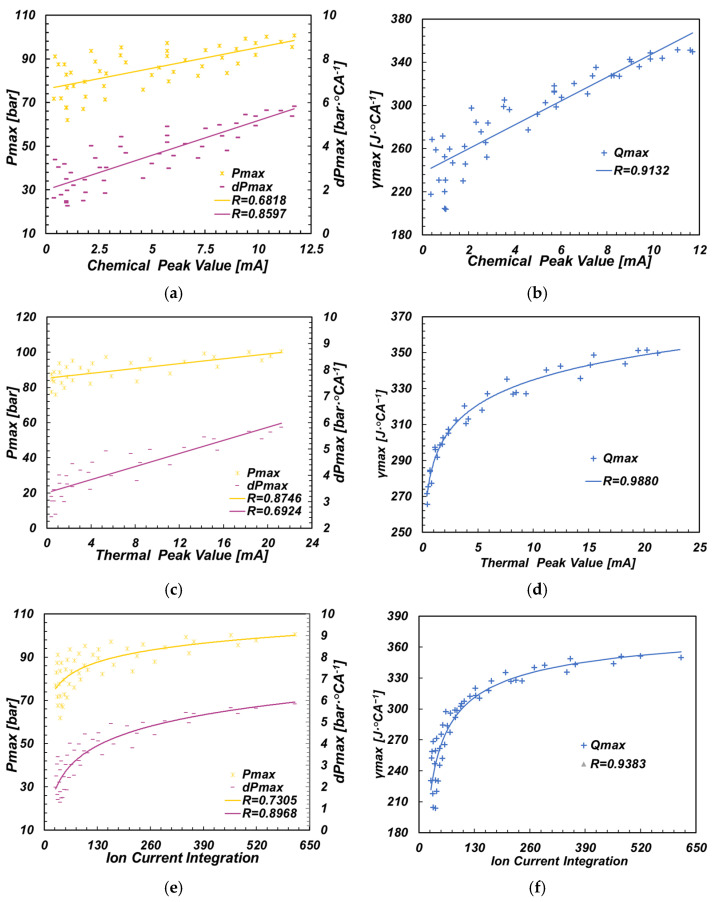
Relationship between ion current amplitudes and Pmax, dPmax and γmax. (**a**) CPV and Pmax; (**b**) CPV and γmax; (**c**) TPV and Pmax; (**d**) TPV and γmax; (**e**) ICI and Pmax; (**f**) ICI and γmax.

**Figure 10 sensors-22-04660-f010:**
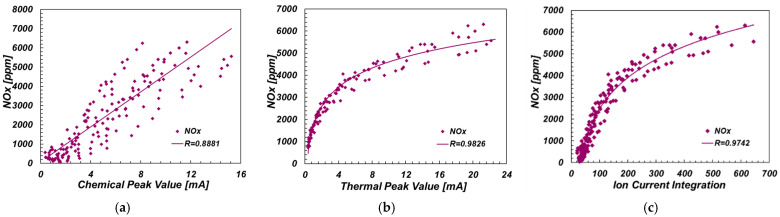
Relationship between ion current amplitudes and NOx. (**a**) NOx and CPV; (**b**) NOx and TPV; (**c**) NOx and ICI.

**Figure 11 sensors-22-04660-f011:**
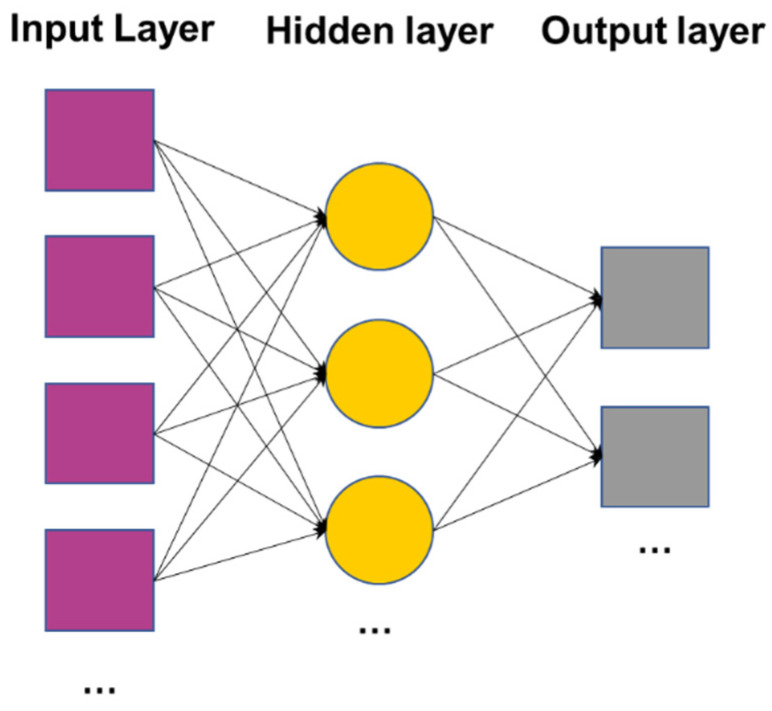
Network topology.

**Figure 12 sensors-22-04660-f012:**
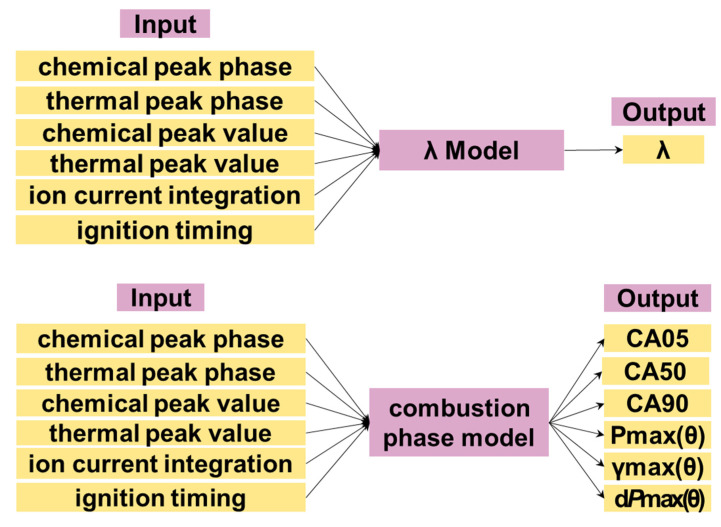
Inputs and outputs of 4 models.

**Figure 13 sensors-22-04660-f013:**
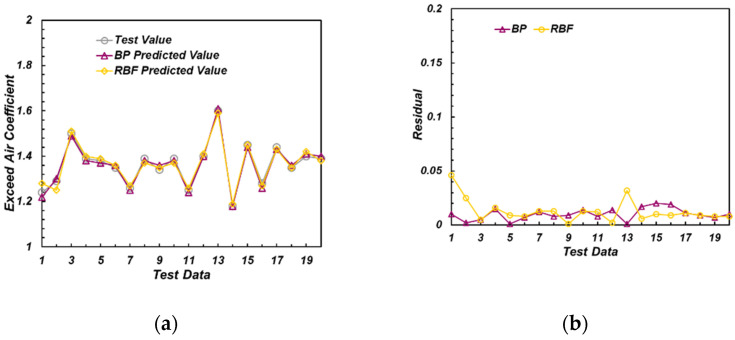
Predicted and experimental value of excess air coefficient (with thermal phase). (**a**) Model results of exceed air coefficient; (**b**) Residual of the model results.

**Figure 14 sensors-22-04660-f014:**
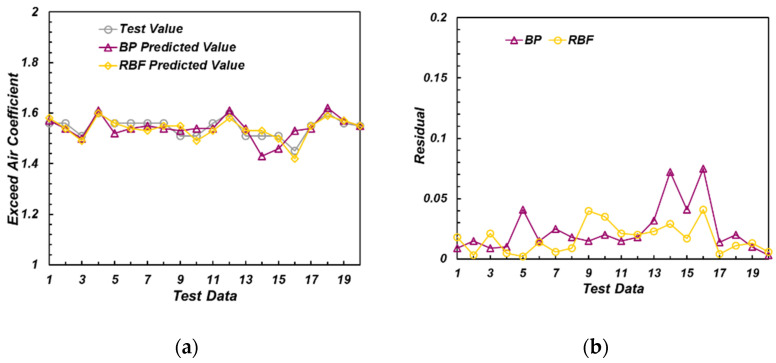
Predicted and real value of excess air coefficient (without thermal phase). (**a**) Model results of exceed air coefficient; (**b**) Residual of the model results.

**Figure 15 sensors-22-04660-f015:**
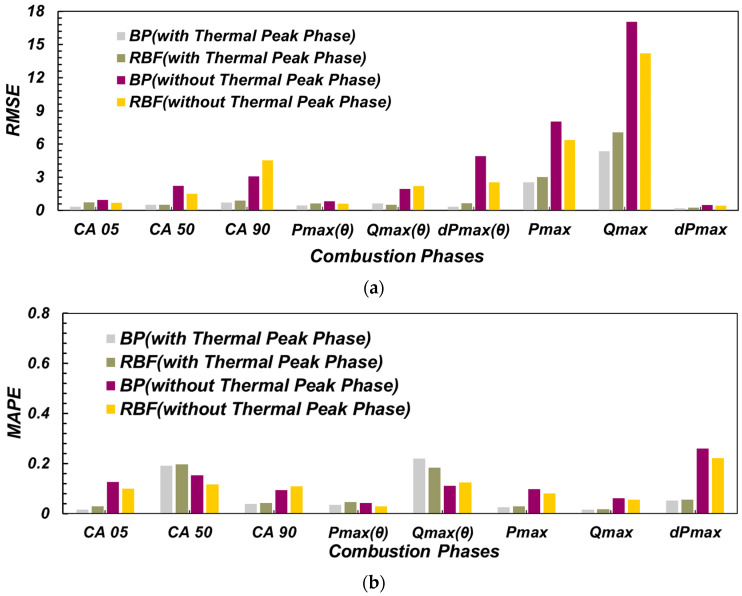
Accuracy of combustion phase model. (**a**) RMSE of the model results; (**b**) MAPE of the model results.

**Figure 16 sensors-22-04660-f016:**
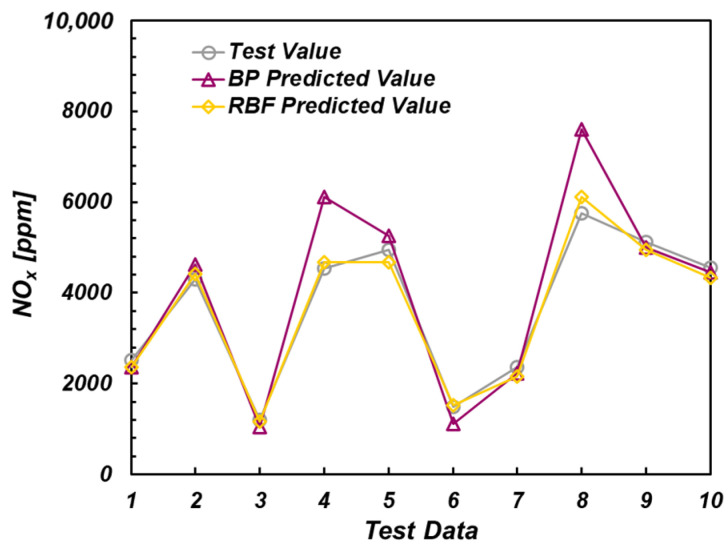
Prediction results of NOx emission values.

**Table 1 sensors-22-04660-t001:** Engine specifications (YC6K).

Parameters	Value
Diameter/mm	129
Stoke/mm	165
Compression ratio	10:1
Fuel injection type	PFI (port fuel injection)
Engine speed/(r·min^−1^)	1500

**Table 2 sensors-22-04660-t002:** Engine specifications (YC6K).

Load	Ignition Timing	λ
10%	25–40 °CA BTDC	1.2–1.6
20%	25–40 °CA BTDC	1.2–1.6
25%	25–40 °CA BTDC	X-0.2–1.6
30%	25–40 °CA BTDC	1.2–1.6
40%	25–40 °CA BTDC	1.2–1.6
50%	25–40 °CA BTDC	X-1.2–1.6
60%	25–40 °CA BTDC	X-0.2–1.6
70%	25–40 °CA BTDC	X-0.2–1.6
75%	25–40 °CA BTDC	X-0.2–1.6
80%	25–40 °CA BTDC	X-0.2–1.6
90%	25–40 °CA BTDC	X-0.2–1.6

**Table 3 sensors-22-04660-t003:** Correlation coefficients between ion current phase parameters and combustion phase parameters.

		CA05	CA50	CA90	Pmax(θ)	γ(θ)	dPmax(θ)
CPP	λ = 1.3	0.9668	0.9658	0.9203	0.9513	0.9563	0.9624
	λ = 1.4	0.9019	0.9136	0.8986	0.8911	0.8999	0.9113
	λ = 1.5	0.9059	0.9066	0.9040	0.9078	0.9008	0.3274
TPP		0.9617	0.9635	0.8764	0.9793	0.9610	0.9793

## Data Availability

Not applicable.

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
