# Peer review of "A Virtual Combustion Sensor Based on Ion Current for Lean-Burn Natural Gas Engine"

_sensors, 2022, doi:10.3390/s22134660_

Round 1
Reviewer 1 Report
In this study, the authors present performance data for a conceptual sensor designed to detect key combustion parameters of a natural gas engine. Four kinds of neural network models are proposed and their predicted parameter values compared against experimental counterparts. The BP model had the highest prediction accuracy of phase and amplitude parameters of combustion, while the RBF model for emission parameter. Overall, the manuscript is fairly well written and with high quality graphics. A few things to consider when performing the revision are listed below with direct reference to location in the manuscript:
Abstract: in general, I would suggest using past tense rather then present tense throughout the whole abstract
Lines 17-18: replace "mentioned above" with "above mentioned" or "aforementioned"
Line 18: "take" (plural form)
Line 21: rephrase to "which feature a thermal phase peak"
Line 128, x-axis graph label: replace "deg" with the symbol "o"
Line 148 : Capital W is not necessary in "Where", which should be written without an indent
Line 151: comma at beginning of this line does not belong here and should be fitted on line 150, at the end
Line 258: replace "conduct" with "determine" or "inflict"
Lines 260-261 and again 364-365: blank spaces needed before 1.6 and 15, respectively
Line 300: Capital W is not necessary in "Where", which should be written without an indent; the "i"s in "mi" and "fi" should appear as subscripts and "N" should not be capitalized (should be little "n") just as they appear in the mathematical formulae above
There appears to be a section 6 of the manuscript "Patents" with no content at all. What is this section about?
Author Response
Dear Editor and Reviewers:
Thank you for your letter and for the reviewers’ comments on our manuscript entitled “A Virtual Combustion Sensor Based on Ion Current for Lean-burn Natural Gas Engine” (Manuscript number: sensors-1777574). These comments are very valuable and helpful to revise and improve our paper. We have studied comments carefully and made corrections which we hope to get your approval. The revised portion of the paper has been marked. The response to the reviewers’ comments are in the attached document.

Reviewer 2 Report
The article describes a soft sensor detecting combustion parameters using the features of the ion current waveform of a marine engine fed with natural gas. In my opinion, the article is interesting, but I have some comments to improve its quality
11. How the Ion Current Integration (ICI) characteristic is calculated? Is this the area under the ionization current curve? If so, within what limits does integration take place? Is it time integral of crank angle integral? Please provide the formula for the ICI
22. An abbreviation of 'crank angle' (probably) appears in various places in the article. For example, in fig. 3 - Crank Angle [deg], in fig. 4,5,7,8 - CAD, in fig. 6 - oCA BTCD, and in table 2 - oCA BTDC. Please explain and standardize the abbreviations
33. Is the order of the approximating function correctly selected in Fig. 9f?
44. Fig.12 a and b: The CPV input parameter occurs twice. Maybe it would be clearer to use the full parameter names, not just abbreviations.
55. Line 291: There is an unclear sentence: Then, 20 samples from 388 samples (single flow thermal phase with ion) and 127 samples (single flow thermal phase without ion) were taken as test and validation samples to verify the accuracy of the established virtual sensor.
Why were only 20 samples used for testing and validation, it should be around 70% for training. What validation was used? This issue should be further explained.
66. Fig.15a: To compare the RMSE calculated for different parameters, the measures would have to be normalized beforehand. The comparison makes sense for the values of relative errors, as in Fig. 15b.
77. Fig.15b: The dPmax(q) parameter is over 20 times greater than the others, which makes the comparison difficult. Many errors are close to zero. Maybe the graph could be shown without dPmax(q)?
88. Why were the BN and RBF neural networks used in the work? How do the authors justify this choice?
99. Have other exhaust components besides NOx been tested, eg CO or HC. Are they correlated in any way with the ionization current? Is it described somewhere in the reading?
Author Response

(The authors gave the same response as above.)
